



# A Convolutional Neural Network for Spatial Downscaling of Satellite-Based Solar-Induced Chlorophyll Fluorescence (SIFnet)

Johannes Gensheimer[1,2], Alexander J. Turner[3], Philipp Köhler[4], Christian Frankenberg[4], and Jia Chen[1]

[1]Environmental Sensing and Modeling, Technical University of Munich (TUM), Munich, Germany
[2]Max Plank Institute for Biogeochemistry, 07745 Jena, Germany
[3]Department of Atmospheric Sciences, University of Washington, Seattle, WA, USA
[4]Division of Geological and Planetary Sciences, California Institute of Technology, Pasadena, CA, USA

**Correspondence:** Johannes Gensheimer (johannes.gensheimer@bgc-jena.mpg.de), Jia Chen (jia.chen@tum.de), Alexander J. Turner (turneraj@uw.edu)

**Abstract.** Gross primary productivity (GPP) is the sum of leaf photosynthesis and represents a crucial component of the global carbon cycle. Space-borne estimates of GPP typically rely on observable quantities that co-vary with GPP such as vegetation indices using reflectance measurements (e.g., NDVI, $NIR_v$, and kNDVI). Recent work has also utilized measurements of solar-induced chlorophyll fluorescence (SIF) as a proxy for GPP. However, these SIF measurements are typically coarse resolution

while many processes influencing GPP occur at fine spatial scales. Here, we develop a Convolutional Neural Network (CNN), named SIFnet, that increases the resolution of SIF from the TROPOspheric Monitoring Instrument (TROPOMI) on board of the satellite Sentinel-5P by a factor of 10 to a spatial resolution of 500 m. SIFnet utilizes coarse SIF observations together with high resolution auxiliary data. The auxiliary data used here may carry information related to GPP and SIF. We use training data from non-US regions between April 2018 until March 2021 and evaluate our CNN over the conterminous United

States (CONUS). We show that SIFnet is able to increase the resolution of TROPOMI SIF by a factor of 10 with a $r^2$ and $RMSE$ metrics of 0.92 and 0.17 mW m$^{-2}$ sr$^{-1}$ nm$^{-1}$, respectively. We further compare SIFnet against a recently developed downscaling approach and evaluate both methods against independent SIF measurements from Orbiting Carbon Observatory 2 and 3 (OCO-2/3). SIFnet performs systematically better than the downscaling approach ($r$ = 0.78 for SIFnet, $r$ = 0.72 for downscaling), indicating that it is picking up on key features related to SIF and GPP. Examination of the feature importance

in the neural network indicates a few key parameters and the spatial regions these parameters matter. Namely, the CNN finds low resolution SIF data to be the most significant parameter with the $NIR_v$ vegetation index as the second most important parameter. $NIR_v$ consistently outperforms the recently proposed $kNDVI$ vegetation index. Advantages and limitations of SIFnet are investigated and presented through a series of case studies across the United States. SIFnet represents a robust method to infer continuous, high spatial resolution SIF data.



## 1 Introduction

Photosynthesis represents the single largest $CO_2$ flux between the atmosphere and the biosphere. At the canopy level, the sum of all leaf photosynthesis is termed Gross Primary Productivity (GPP) and accurate characterization of GPP represents a major uncertainty in the carbon cycle (Friedlingstein et al., 2019). Directly measuring GPP from remote sensing systems (e.g.,

satellites) is not presently possible. Instead, previous work has utilized stationary measurements of net ecosystem exchange (NEE) from flux towers that can be decomposed into GPP and respiration (e.g., Reichstein et al., 2005). Observable quantities from satellites (e.g., vegetation indices computed from reflectance data) are then related to GPP inferred from flux towers (e.g., Huete et al., 2006; Jung et al., 2019; Sims et al., 2006; Zeng et al., 2020) in light use efficiency (Mahadevan et al., 2008) or machine learning models Jung et al. (2019) to derive global estimates of GPP.

Vegetation indices such as NDVI and $NIR_v$ combine two (or more) spectral bands with different absorption characteristics (Hanes, 2013) to infer quantities related to plant physiology and canopy structure. The MODIS instrument was launched on the Terra and Aqua satellites in 1999 and 2002. This instrument has proved particularly useful due, in part, to the long operational lifetime. More recently launched satellites, like Sentinel-5P, carry instruments with the necessary signal to noise ratio and spectral resolution to retrieve solar-induced chlorophyll fluorescence (SIF), which is a measure of re-emitted photons

by the chlorophyll during photosynthesis (Köhler et al., 2018). Vegetation indices (also termed greenness) can be regarded as a measure of photosynthetic capacity whereas SIF indicates photosynthetic activity (Sellers, 1985). SIF has been shown to be a powerful proxy for estimating GPP (Magney et al., 2019; Turner et al., 2021), to capture the impact of drought on photosynthetic activities across different vegetation types (Shekhar et al., 2020a; Castro et al., 2020), and to assess the regional source of carbon emissions (Shekhar et al., 2020b).

Köhler et al. (2018) described the first retrievals of SIF from TROPOMI, the sole instrument on the Sentinel-5P satellite. The TROPOMI instrument has an equatorial crossing time of 13:30 local solar time (LST) and a 16 day orbit cycle. TROPOMI has a wide swath (2600 km across track) that allows for near-daily temporal resolution and a spatial resolution of 5.5x3.5 km. This was a substantial improvent to previous satellite instruments measuring SIF that were limited to $40 \times 40$ $km^2$ spatial resolution (Joiner et al., 2013). Despite the improves spatial resolution of TROPOMI, there have been efforts to estimate SIF at

finer spatial scales (e.g., Turner et al., 2020). This is motivated by the importance of fine-scale phenomena in the carbon cycle such as ecosystem fragmentation (e.g., Haddad et al., 2015).

Globally, 20% and 70% of the remaining forests are within a distance of 100 m and 1 km, respectively, from the forest edges, meaning that most of the forests are fragmented (Haddad et al., 2015). Reinmann and Hutyra (2017) show that the carbon uptake and storage of trees near the forest edge increase up to 13±3% and 10±1%, respectively. On the other hand most of

our understanding about forest carbon fluxes comes from intact ecosystems, resulting in a mismatch between the ecosystems we are trying to quantify and the data we are using to do so (Smith et al., 2018). Higher resolution estimates of photosynthetic activity might enable to include fragmentation effects of ecosystems to global carbon cycle estimates or biosphere models like, Jung et al. (2019); Wu et al. (2021); Turner et al. (2021). Additionally, recent work has shown the importance of fine-scale variations in the urban biosphere on the overall carbon flux for a city (e.g., Miller et al., 2020).



There has been some recent work with the goal of increasing the resolution of existing global SIF estimates through down-scaling methods (i.e., physics based methods). For example, Turner et al. (2020, 2021) used $NIR_v$ to partition SIF within a particlar TROPOMI scene and oversampled it using a 16 day window afterwards, resulting in a daily 500m SIF estimate over CONUS. Duveiller et al. (2020) downscaled GOME-2 satellite SIF from $0.5°$ to $0.05°$ using a parameterization with a term for the fraction of absorbed photosynthetically active radiation (fAPAR), one for water stress, and one for heat stress based on
MODIS data. Siegmann et al. (2021) used airborne data to downscale far-red SIF from canopy to leaf level.

    Machine learning has also been used to create global, high resolution SIF data sets. Li and Xiao (2019), Yu et al. (2019), and Zhang et al. (2018) used spectral bands from MODIS as input to neural networks that were trained with OCO-2 SIF data to build global continuous SIF products at $0.05°$ resolution. OCO-2 has a narrow swath and, therefore, the networks are trained only in the regions where OCO-2 SIF is available by using MODIS data as input. After training, the global MODIS
data is used as input to estimate SIF on a global scale. Gentine and Alemohammad (2018) uses MODIS reflectance data as input and predicts GOME-2 normalized by clear-sky irradiance. Multiplying that with a MODIS derived photosynthetic active radiation product results in a MODIS only estimated SIF, termed RSIF. Zhang et al. (2021) trains a CNN with MODIS data on the artificial GOSIF data set (Li and Xiao, 2019) at a resolution of $0.05°$ and uses the trained network and MODIS data at a resolution of $0.008°$ to estimate SIF at $0.008°$. The physics based downscaling approach from Turner et al. (2020) can only
consider one variable for weighting the SIF signal while the machine learning based approaches in the literature can consider more than one variable – but many do not use SIF data as an input to their model, meaning that they estimate SIF based on reflectance data.

    Here we build a convolutional neural network to obtain high-resolution SIF, named SIFnet. SIFnet increases the spatial resolution of TROPOMI SIF by considering coarse resolution SIF together with high resolution auxiliary data as input. This
auxiliary data consists of either proxies of SIF or photosynthetic drivers. SIFnet is trained using data with near-global coverage. Different model parameters (structure, input features, and scaling factors) are compared and evaluated. After training the model, the resolution of TROPOMI SIF is refined by a factor of ten to a spatial resolution of $0.005°$. This product is then compared against a recent downscaling method from literature (Turner et al., 2020). Both high resolution estimates are validated over CONUS against the independent SIF measurements of the OCO-2 and 3 instruments (OCO-2 and OCO-3, respectively).
(OCO-2 Science Team/Michael Gunson, 2020; OCO-3 Science Team/Michael Gunson, 2020).

## 2   Datasets

### 2.1   Data sources

The input data to the neural network is listed in Table 1. These diverse global data products are expected to capture a broad
range of photosynthetic drivers. The table divides the data into time-varying or time-invariant and training or validation. The native spatial resolution is shown in the last column of Table 1. In a first step, all data sets are aggregated to $0.05°$ spatial resolution and 16 day time steps. In case of a higher native spatial resolution, the data is regridded by computing the mean





value that falls into the coarse resolution grid cell. In case of coarser resolutions than $0.05°$ it is resampled to the common grid. Quality control flags and cloud filtering are applied when necessary.

MODIS measures the reflected radiance from the earth surface in seven different spectral bands covering the visible and infrared spectral region. Vegetation indices are computed by combining the near-infrared (where chlorophyll is non-absorbing) and the red band (where chlorophyll is highly absorbing) (Hanes, 2013). Specifically, the normalized difference vegetation index (NDVI) (Tucker, 1979), near-infrared vegetation index (NIR$_v$) (Badgley et al., 2017), the kernel NDVI (kNDVI) (Camps-Valls et al., 2021), and the enhanced vegetation index (EVI) (Huete et al., 2002) are computed as:

$$NDVI = \frac{\rho_{NIR} - \rho_{RED}}{\rho_{NIR} + \rho_{RED}} \tag{1}$$

$$NIR_v = \rho_{NIR} \cdot NDVI \tag{2}$$

$$kNDVI = tanh(NDVI^2) \tag{3}$$

$$EVI = G \cdot \frac{\rho_{NIR} - \rho_{RED}}{\rho_{NIR} + C_1 \cdot \rho_{RED} - C_2 \cdot \rho_{BLUE} + L} \tag{4}$$

EVI coefficients for MODIS are: L=1, C$_1$=5, C$_2$=7.5, G=2.5 (Huete et al., 2002). $\rho_{NIR}$ is the near infrared band, $\rho_{RED}$ is the red band, $\rho_{BLUE}$ is the blue band from the MODIS satellites.

    Temperature and precipitation is taken from ERA5-Land data at the time step of interest and with a delay of one time step.
Soil moisture has been shown to be a strong driver of global photosynthesis due, in part, to its impact on vapor-pressure deficit (Humphrey et al., 2021). Here we use the coarse resolution NASA USDA SMAP soil moisture (Entekhabi et al., 2010) as a model input and explore its correlation with TROPOMI SIF. The cosine of the SZA is a proxy for photosynthetic active radiation (PAR) under cloud-free conditions (Chen et al., 2020; Turner et al., 2021).

Time invariant data sets consist of elevation data, fractional land cover classification, and forest fragmentation data. The land
cover classification (Buchhorn et al., 2020) is resampled to 11 fractional classes. The forest fragmentation data consists of two channels and has a native resolution of 30 m. One channel describes the share of forest within the grid cell (forest share) and the other how much of that forest is edge forest (defined as a maximum distance to an edge or other land cover type of 30 m). OCO-2 and OCO-3 have high spatial resolution (2.25 km x 1.29 km) but small swaths (10 km) and a 16-day revisit time.



**Table 1. Data sets used in this work.**

| Data | | time invariant | training | validation | spatial resolution |
|---|---|---|---|---|---|
| Sentinel-5P TROPOMI[1] | SIF@ 740 nm | | X | | 0.05° |
| MODIS MCD43A4.v006 (v06)[2] | MODIS bands | NIR, RED, BLUE, GREEN, SWIR1, SWIR2, SWIR3 | | X | | 500m |
| | Vegetation Indices | NIRv, kNDVI, NDVI, EVI | | X | | 500m |
| ERA5-Land Hourly - ECMWF Climate Reanalysis[3] | temperature | mean air temperature, mean air temperature with 16 days delay | | X | | 0.1° |
| | precipitation | total precipitation, total precipitation with 16 days delay | | X | | 0.1° |
| NASA USDA Enhanced SMAP Soil Moisture[4] | surface soil moisture, sub-surface soil moisture | | X | | 10 km |
| Solar zenith angle[5] | cosine of the solar zenith angle | | X | | computed |
| USDA GMTED2010: Global Multi-resolution Terrain Elevation Data 2010[6] | elevation | X | X | | 7.5 arc seconds |
| Copernicus Corine global land cover classification (CLC2018)[7] | Non-vegetated, ENF, EBF, DNF, DBF, MF, UF, shrubs, grassl., crops,wetland | X | X | | 100m |
| Forest fragmentation[8] | forest share | X | X | | 30m |
| | edge share | X | X | | 30m |
| OCO-2[9] | SIF @ 740 nm | | | X | 2.25 x 1.29 km |
| OCO-3[10] | SIF @ 740 nm | | | X | 2.25 x 1.29 km |

References: [1] Köhler et al. (2018); [2] Schaaf, C., Wang, Z. (2015); [3] ERA (2017); [4] Entekhabi et al. (2010); [5] PySolar (2021); [6] Danielson and Gesch (2011); [7] Buchhorn et al. (2020); [8] Morreale et al. (2021); [9] OCO-2 Science Team/Michael Gunson (2020); [10] OCO-3 Science Team/Michael Gunson (2020)



## 2.2 Covariation of input datasets with SIF



**Figure 1. Scatter comparison of SIF to timely changing auxiliary data.** The time span of measurements is from April 2018 to March 2021 in 16-day resolution. Longitude and latitude borders are from -180° to 180° and -60° to 70°, respectively. The comparison resolution corresponds to the lowest resolution of the two corresponding products. For all MODIS data the resolution is 0.05° and for precipitation, air temperature, surface soil moisture (ssm), and subsurface soil moisture (susm) 0.1°.

We are interested in understanding what these different data sets are telling us about SIF and also how they co-vary with each other. We compare all collected time variant data against TROPOMI SIF in the spatial and temporal domain. As a quantitative measure, we compute the Pearson correlation coefficient ($r$) (Benesty et al., 2009). Figure 1 shows a scatter comparison of





$SIF$ against the auxiliary data at the lowest resolution of the two corresponding sets. Negative $SIF$ values (on the x-axis in Figure 1) are due to relatively high retrieval errors which scale with radiance levels (Köhler et al., 2018).

In Figure 2 shows spatial patterns of the Pearson correlation coefficients between both $NIR_v$ and $kNDVI$ to $SIF$. In both our spatial (Fig. 2 and temporal (Fig. 1) analyses, we find $NIR_v$ is a better predictor for SIF than $kNDVI$, which contradicts the recent findings from Camps-Valls et al. (2021). However, Camps-Valls et al. (2021) used GOME-2 SIF instead of TROPOMI SIF.

The vegetation index $NIR_v$ outperforms $kNDVI$ in nearly all vegetated regions. Only central Asia, Sahara, and very high
latitudes show a better correlation of $kNDVI$ with SIF. At the same time, these regions generally show a weaker correlation of vegetation indices with SIF.

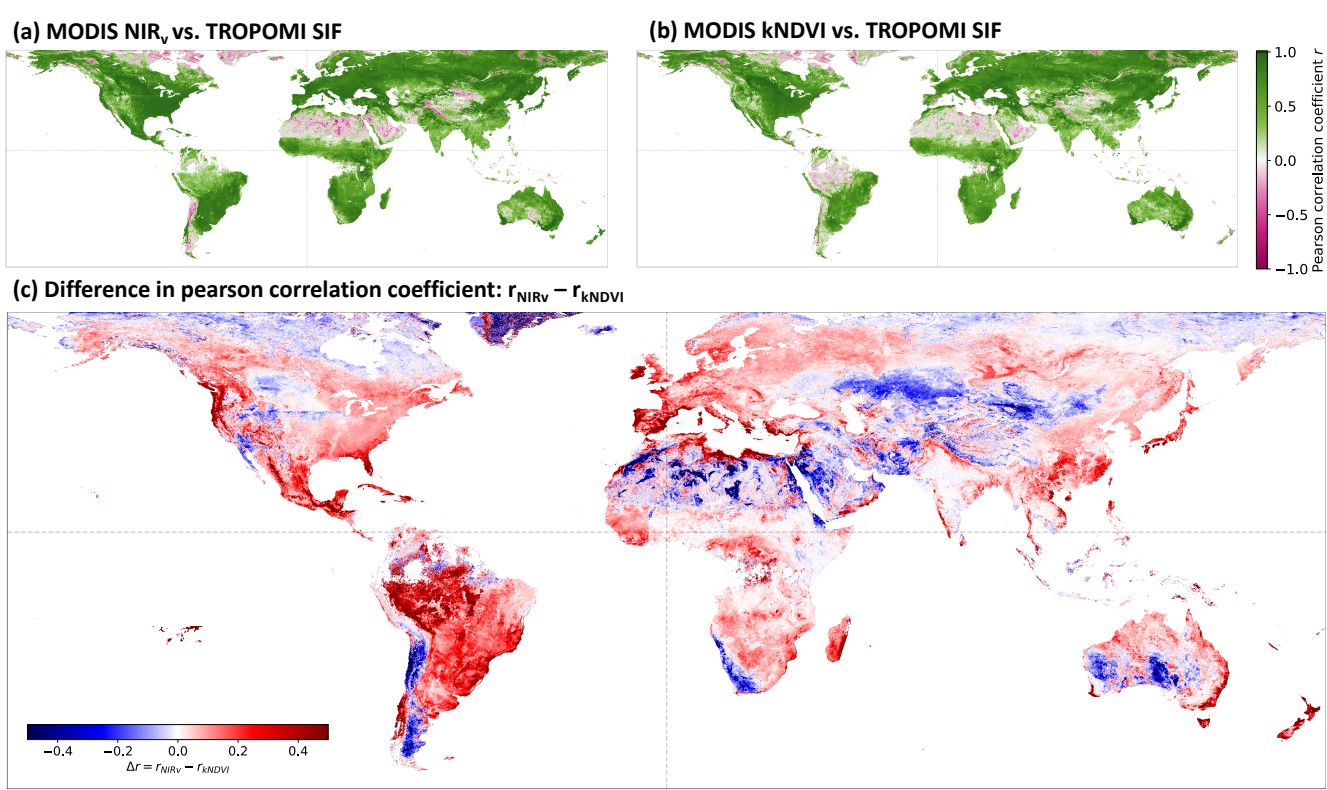

**Figure 2. Pearson correlation coefficient of NIR_v and kNDVI to TROPOMI SIF.** Data is compared at 0.05° spatial resolution and in 16-day time steps starting in April 2018 until March 2021. The value per grid cell in a) and b) represents the Pearson correlation coefficient of the vegetation index to $SIF$ in time. Panel c) represents the difference in correlation of the vegetation indices to $SIF$.

Figure 1 shows several features that are highly correlated to SIF (e.g., $NIR_v$ and $kNDVI$), suggesting a collinearity between features. Using principal component analysis (PCA), we find that the first 9 PCs (of in total 19) represent 99% of the variance in the time-varying datasets and that 13 (of in total 15) of the PCs represent 99% of the variance in the time-invariant
datasets. There are many instances where PCs are clearly interpretable and NNs can handle collinearity, in contrast to e.g.





linear regression methods (De Veaux and Ungar, 1994). We therefore keep all input features in our investigation and expect a longer convergence time during training of the CNN.

# 3 Development and Optimization of SIFnet



**Figure 3. CNN model structure and training/estimation method.** Yellow and red blocks are convolutional and ReLU layers, respectively. Notation of convolutional layers: k(X1,X2): kernel sizes are X1,X2, chY: number of channels is Y. For training, the data is upscaled. We input auxiliary data at the target resolution and SIF data with a factor of 10 coarser.





## 3.1 Training and Optimization of the Neural Network

Convolutional neural networks (CNNs) are supervised machine learning methods that need matching feature and ground truth data pairs to compute the loss that is back propagated. As such, we being by coarsening SIF data to 0.5° and used with auxiliary data at 0.05° as input to SIFnet, allowing us to estimate SIF at 0.05°. The model output is compared against the measured TROPOMI SIF at 0.05°. After optimizing the model it can resolve a scaling factor of ten between coarse resolution input SIF and model output SIF. Figure 3 visualizes this method. In the following step of estimating high resolution SIF, the

feature SIF data has a resolution of 0.05° and auxiliary data of 0.005°, resulting in a model output of SIF at 0.005°.

Figure 3 shows our chosen CNN model structure for SIFnet. The model consists of convolutional and Rectified Linear Units (ReLU) layers that are arranged in a sequence. After the first convolutional block there is a residual connection that skips 2 convolutional and one ReLU layer. Convolutional kernel sizes are either (3,3) or (1,1). This structure is adapted from literature findings from e.g. Lim et al. (2017). Further, several model structures (Supplemental Section S4.3) with a different amount of

layers, channels or residual blocks are compared. The chosen model structure represents the best trade-off between complexity and performance. The input feature collinearity and PCA presented in Supplemental Section S3 show that some input features have high correlations with each other. Nine out of the 19 PCs in the time variant and 13 out of the 15 PCs in the time invariant data carry above 99% of the variance. This suggests to use fewer channels in the CNN layers than the feature dimension (because some variables are similar). Therefore the number of channels in the first layer of SIFnet reduces the complexity from

34 to 16 channels (Figure 3). More complex model structures did not result in a notably improved loss metrics (Supplemental Section S4.3).

For training SIFnet we use three years of data (April 2018 - March 2021) in 16 day time steps. The study regions are shown in Supplemental Section S2. There are five folds used as training data: two folds over Asia, one over Europe, one over southern part of Africa, one over South America. Our validation region is North America (Figure S4). The hyperparameter tuning is

done by training the model on the five folds and computing the loss of the validation data. The parameters are optimized to minimize the loss of the validation data set. Due to computational reasons and the size of the data set we do not apply a cross validation in the optimization process. The final product consists of high resolution SIF at 0.005° and is validated against independent SIF measurements of the instruments OCO-2 and OCO-3.

We center and scale each feature individually by subtracting the mean and normalizing by the standard deviation. For data

augmentation of the training data, we use random crops and random flips. Each day of one fold has a matrix size of 1200x900 pixels. We are analyze 69 days in 16 day steps over 3 years. For each input during the training process we randomly crop a matrix with a size of 100x100 pixels. As some areas have a large fraction of missing values (e.g., due to water or clouds), we only use cropped matrices that consist of >80% valid pixels in the SIF product. Further, we randomly flip vertically and horizontally, both with a probability of 0.5. These data augmentation methods provide us a huge database that should avoid

overfitting the network parameters. During training water bodies and all missing values are set to zero.

Our individual loss function is comprised of two loss terms. We use the Mean Squared Error (MSE) loss in combination with the Structural Dissimilarity Index (DSSIM). The DSSIM is the counter-measure of the Structural Similarity Index (SSIM):





DSSIM = 1-SSIM (Brunet et al., 2011). Therefore, we are not only optimizing the overall deviation of the estimated SIF to the measured SIF but also on structural patterns. Supplemental Section S4.4 shows the benefit of including both MSE and SSIM

terms in the loss function. Equation 5 shows our loss function:

$$\mathcal{L} = a \cdot MSE + b \cdot DSSIM = a \cdot \frac{1}{n} \cdot \sum_{i=1}^{n}(y_i - \widetilde{y_i}) + b \cdot (1 - \frac{(2 \cdot \mu_Y \cdot \mu_{\widetilde{Y}} \cdot (2 \cdot \sigma_{Y\widetilde{Y}} + c_2)}{(\mu_Y^2 + \mu_{\widetilde{Y}}^2 + c_1) \cdot (\sigma_Y^2 \cdot \sigma_{\widetilde{Y}}^2 + c_2)}) \tag{5}$$

where $n$ is the number of datapoints; $y_i$ is the datapoint $i$ in measured (ground truth) SIF; $\widetilde{y_i}$: is the datapoint $i$ in estimated SIF; $Y$ are all datapoints of measured (ground truth) SIF; $\widetilde{Y}$ are all datapoints of estimated SIF; $\mu_Y$ is the mean of $Y$; $\mu_{\widetilde{Y}}$ is the mean of $\widetilde{Y}$; $\sigma_Y$ is the variance of $Y$; $\sigma_{\widetilde{Y}}$ is the variance of $\widetilde{Y}$; $\sigma_{Y\widetilde{Y}}$ is the covariance of $Y$ and $\widetilde{Y}$; $c_1 = (k_1 L)^2$ and $c_2 = (k_2 L)^2$

being variables for stabilisation with $L = 2^{bitsperpixel} - 1$, $k_1 = 0.01$ and $k_2 = 0.03$. The parameters $a$ and $b$ define the weights on the overall loss of the two individual losses. The overall model performance did not show a notable sensitivity to different $a$ and $b$ values. To approximately keep the individual losses in the same order of magnitude we set $a = 1$ and $b = 0.3$. DSSIM is in the range of 0 to 1, with 0 meaning structural similar and 1 structural dissimilar.

We use the optuna library for for hyperparameter tuning of the learning rate, weight decay, and epoch of the CNN (Akiba

et al., 2019). Here, a Tree-structured Parzen Estimator Sampler suggest the parameters of the next trial which is based on a Gaussian Mixture Model. Supplemental Section S4.1 provides more details on this hyperparameter tuning.

### 3.2  Results of Model Optimization

Figure 4 summarizes the results of the optimized model. We observe an overall $r^2$ of 0.92, $SSIM$ of 0.87, and $RMSE$ of 0.17 $mWm^{-2}sr^{-1}nm^{-1}$ between the estimated SIF from SIFnet and retrieved SIF from TROPOMI at 0.05° (Fig 4e). $SSIM$

is calculated by comparing the average SIF signal of the three years under investigation. Figure 4e shows the three metrics for each month of the year. We observe the lowest $r^2$ values in January, February, and March. These are associated with low SIF values and, consequently, lower signal to noise ratios which drive the decreased performance. $SSIM$ also indicates reduced performance during this time period. $RMSE$ values are correlated with overall productivity with the lowest RMSE in winter; this is expected as this metric depends on the magnitude of the signal.

### 3.3  Which features drive SIFnet?

We are particularly interested in understanding which features drive our neural net. Here we evaluate the feature importance using the Permutation Feature importance method (Breiman, 2001; Fisher et al., 2019; Gregorutti et al., 2015, 2017) with our North American validation data at a target resolution of 0.05°. The method first computes the RMSE including all input features (RMSE$_{orig.}$). We then apply the following three steps:

1. Shuffle all pixels of one input feature randomly in time and space.

2. Compute the new RMSE of the estimation (RMSE$_{F,shuf.}$).

3. Compare the shuffled RMSE to the original: $d_{F,shuf.} = RMSE_{F,shuf.}/RMSE_{orig.}$


**(a) Model Input: Upscaled TROPOMI SIF at 0.5°**

**(b) Model Estimate: SIFnet SIF at 0.05°**

**(c) Validation data: TROPOMI SIF at 0.05°**

**(d) Scatter comparison of TROPOMI and CNN SIF**

$r^2 = 0.92$

$RMSE = 0.17$

$SSIM = 0.87$

**(e) Metrics dependent on time**

04/18-03/19    04/19-03/20    04/20-03/21

**Figure 4. Test set results of CNN training at 0.05°.** (a) shows low resolution SIF that is used as model input. (b) shows the estimated SIF at 0.05° by SIFnet. (c) shows the measured TROPOMI SIF at 0.05° from Köhler et al. (2018). (d) shows the scatter comparison between TROPOMI SIF and the SIFnet estimate at 0.05°. (e) shows for each investigated month the metrics $r^2$, $SSIM$, and $RMSE$. Metrics is calculated in 16 day resolution and averaged to monthly values afterwards.



Figure 5 shows the feature importance of clustered input classes and individual features to the overall estimation. Multiple applications of the feature permutation did yielded negligible differences in feature importance.. Figure 5a shows the $RMSE$

share of shuffled data to the $RMSE$ of unshuffled data. SIFnet finds low resolution SIF ($SIF_{LR}$) to be the most important input variable, followed by the vegetation index $NIR_v$ and the cosine of the solar zenith angle ($cos(SZA)$). All other variables do not contribute notably to the model output. This result strengthens our findings from Figure 1 and Figure 2 that $NIR_v$ is better correlated with SIF than $kNDVI$. Further, our feature importance is in line with Dechant et al. (2022) where they find a high correlation of SIF with $NIR_v$ multiplied with Photosynthetic Active Radiation (PAR), of which the $cos(SZA)$ can be

used as a proxy. Figure 5b shows the spatial feature importance over the validation set in North America for the four most important features. We observe that $SIF_{LR}$ has the biggest impact in the Eastern US, which corresponds strongly to the high vs low productivity regions in the US. $NIR_v$ is a strong predictor in Southeastern US and in shrub regions in the Western US. The contribution of $cos(SZA)$ is highest at high latitudes and weakens at lower latitudes. $NIR_v$ is found to be less predictive of SIF at high latitudes. The land mask is the fourth most important input feature and contributes most in shrub regions. These four

features consistently stand out as the strongest predictors. Other inputs such as fragmentation and soil moisture were not found to be strong predictors here. In supplemental Section S4.6 we test higher scaling factors between low and high resolution SIF. Even with scaling factors of 20 and 50 low resolution SIF stays the most and second most important input feature, respectively.

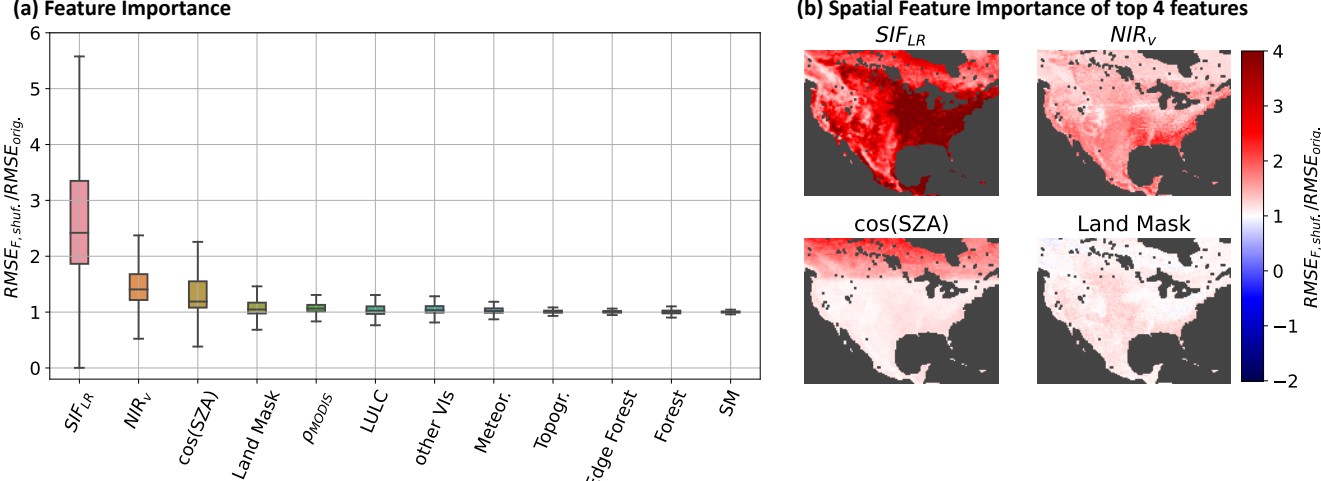

**Figure 5. Feature Importance.** (a) shows the total RMSE of the permuted feature divided by the RMSE without feature permutation (b) shows the RMSE of each pixel with permuted features divided by the RMSE without feature permutation. Some input variables are clustered and all variables of that class are permuted at the same time. $\rho_{MODIS}$: all 7 MODIS bands; LULC: all 11 land cover classes; other VIs: kNDVI, NDVI, EVI; Mereor.: temperature, precipitation, temperature with 16 day delay, precipitation with 16 days delay; SM: surface soil moisture and sub-surface soil moisture.

We also examined different combinations of inputs such as directly including the MODIS bands, as opposed to vegetation indices derived from MODIS bands (see Supplemental Figure S11). Low resolution SIF remains the most important feature,





followed by the NIR band $\rho_{NIR}$. The land cover products increase in relevance. Interestingly, when low resolution SIF is omitted as input for the model, we observe contrasting results to Figure 5 where NIR$_v$ is no longer a leading predictor. We find that $cos(SZA)$, $\rho_{NIR}$, $kNDVI$, and $NDVI$ are the four most important features in this case (see Supplemental Figure S12). This may result from the collinearity between input features. This finding was robust to multiple optimizations and permutations.

### 3.4 Comparison of SIFnet to Downscaled SIF

Figure 6 shows the 0.005° SIF estimated by SIFnet and SIF downscaled SIF from Turner et al. (2020). The difference between the two SIF estimates can be seen in Fig. 6c. SIFnet predicts lower SIF in the western US drylands and higher SIF over forested regions in the eastern US. This prediction of lower SIF in drylands is interesting because Turner et al. (2020) resorted to an ad hoc bias correction in these regions due to a low signal-to-noise ratio. Recent work from Wang et al. (2021) found SIF to perform poorly in the western US drylands. We also observe systematic differences in the predicted SIF in urban areas.

These regions are further evaluated in Supplemental Figure S17. Notably, the SIFnet estimate is systematically lower than the downscaling estimate in most urban regions examined here, Seattle being a notable exception. Both SIFnet and the downscaling approach do allocate SIF to large urban parks and greenspaces, but SIFnet predicts little-to-no SIF over the rest of the urban area. In particular, SIFnet estimates nearly zero SIF in the urban core of Los Angeles and San Francisco. SIFnet and the downscaling method predict comparable SIF as we move away from the urban core. Fine-scale features in the urban region are

visible in both SIF estimates such as the Schiller Woods in Chicago (87.8°W, 42.0°N).

### 4 Validation against OCO-2/3 SIF

The differences in SIF predicted from SIFnet and the downscaled SIF beg the question, *"which is correct?"* Here we evaluate both SIF products against independent SIF observations from OCO-2 and OCO-3(OCO-2 Science Team/Michael Gunson, 2020; OCO-3 Science Team/Michael Gunson, 2020). These instruments have higher spatial resolution than TROPOMI and, as

such, can be used to evaluate the high-resolution patterns predicted by both SIFnet and the downscaling approach. Specifically, OCO-2 and OCO-3 have nadir footprint sizes of 2.25 km x 1.29 km. However, OCO-2 and OCO-3 do not provide full spatial coverage. They observe narrow swaths that are ∼10 km across-track. OCO-3 also provides also a scanning mode to observe urban areas. Here, we use quality-checked OCO-2 data from April 2018 until March 2021 and OCO-3 data from July 2019 until March 2021. To compare the ungridded OCO-2 and OCO-3 data against the SIF estimated from TROPOMI, we compute

the weighted average of all 0.005° grid cells that fall within the bounds of an OCO footprint. Here, the TROPOMI estimates are subsampled to 0.0005° (appr. 50 m at equator) and the mean value is computed for all values which fall into the OCO footprint. For a quantitative comparison between OCO-2/3 and the SIFnet and downscaled estimate the metrics $r$, $r^2$, and $RMSE$ are computed.

The high resolution SIF estimates from SIFnet, the downscaling, and OCO are instantaneous SIF measurements taken at a

specific time of the day, while the time of TROPOMI observations can differ substantially. Here, we compute the daily average



**(a) SIFnet estimated SIF (SIF$_{SIFnet}$) at 0.005°**

**(b) Downscaled SIF (SIF$_{DS}$) at 0.005°**

**(c) Difference between SIFnet and downscaled SIF (SIF$_{SIFnet}$ − SIF$_{DS}$)**

**Figure 6. SIFnet estimated SIF at 0.005° for CONUS and its comparison to downscaled SIF.** (a) shows the SIFnet estimated SIF at 0.005°. (b) shows the downscaled SIF from Turner et al. (2020). (c) shows the difference between SIFnet and downscaled SIF. Negative values imply a higher SIFnet SIF, positive values a higher downscaled SIF value.



SIF by scaling with the cosine of the SZA (Frankenberg et al., 2011):

$$Daily\,SIF(x,y) = SIF(\tau_s,x,y) \cdot \frac{\int_{\tau_o}^{\tau_f} cos[SZA(\tau_s,x,y)]d\tau}{cos[SZA(\tau_s,x,y)]} \tag{6}$$

where $Daily\,SIF(x,y)$ is the daily integrated SIF estimate, $SIF(\tau_s,x,y)$ is the instantaneous SIF at the individual measurement time, $SZA$ is the solar zenith angle, $\tau_s$ is the time of the satellite measurement, $\tau_o$ is the time of sunrise, $\tau_f$ is

the time of sunset. This implicitly assumes that both PAR and SIF scale with cos(SZA) under cloud-free conditions and we neglect Rayleigh scattering as well as gas absorption. Although this approach neglects several water or light conditions, it provides our best estimate of daily SIF and enables comparison between multiple SIF products with different measurement times (Turner et al., 2021; Köhler et al., 2018; Frankenberg et al., 2011). The method is equivalent to the daily correction scheme for OCO-2, OCO-3, and TROPOMI(OCO-2 Science Team/Michael Gunson, 2020; OCO-3 Science Team/Michael Gunson,

2020; Frankenberg et al., 2011). Additionally, we performed a sensitivity study where we trained SIFnet using daily-corrected SIF and found the results to be generally insensitive to the use of instantaneous vs daily-corrected SIF (see Supplemental Figure S18). Following this, we chose to apply the daily correction after deriving the high-resolution SIF .

Figure 7 shows a comparison of both SIFnet and the downscaled SIF to OCO-2 and OCO-3. Specifically, Fig 7a shows the correlation of SIFnet and the downscaled estimate with OCO-2/3 for every 1° pixel over CONUS. Both SIFnet and the

downscaled SIF generally show good agreement with $r$ in excess of 0.7 for most of the high-productivity regions. We observe weaker correlations in the Western drylands due, in part, to a lower signal-to-noise ratio. Overall, we find SIFnet to perform systematically better than the downscaled SIF, as shown in the difference plot. Fig. 7b summarizes these spatial patterns in a scatterplot comparison. SIFnet again shows better performance than the downscaled SIF against OCO-2, OCO-3, and OCO-2/3. The Pearson correlation coefficient $r$ is 0.78 and 0.72 for the SIFnet and downscaled estimate, respectively, when comparing

to all OCO data (right column in Figure 7b). The generally high $RMSE$ indicates different scales and variability in the data sets. These also appear in the comparison of TROPOMI SIF at 0.05° resolution against OCO-2/3 (Supplemental Figure S19). However, the errors generally increase at 0.005 ° compared to the 0.05° resolution.



**(a) Spatial comparison of the correlation of SIFnet and downscaled SIF to OCO-2 and OCO-3**

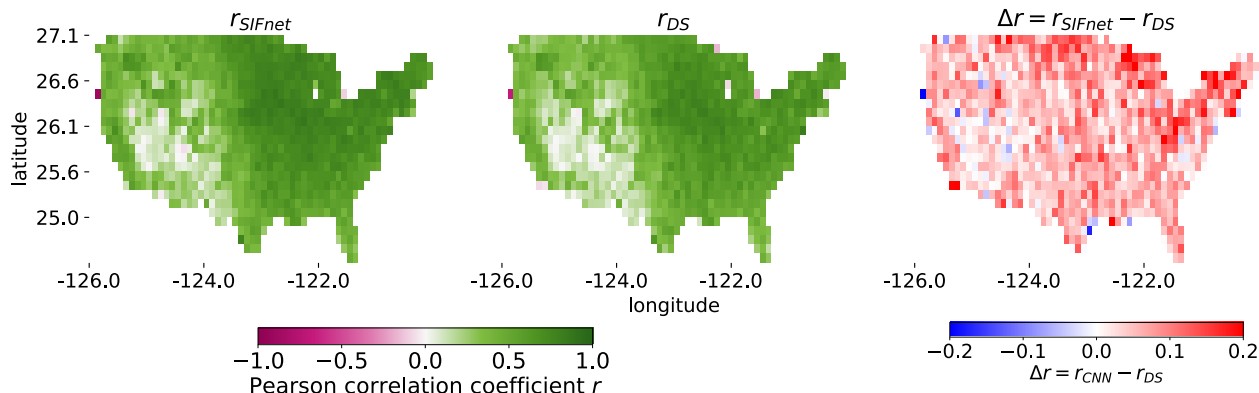

**(b) Validation of SIFnet and downscaled SIF to OCO-2 and OCO-3 over CONUS**

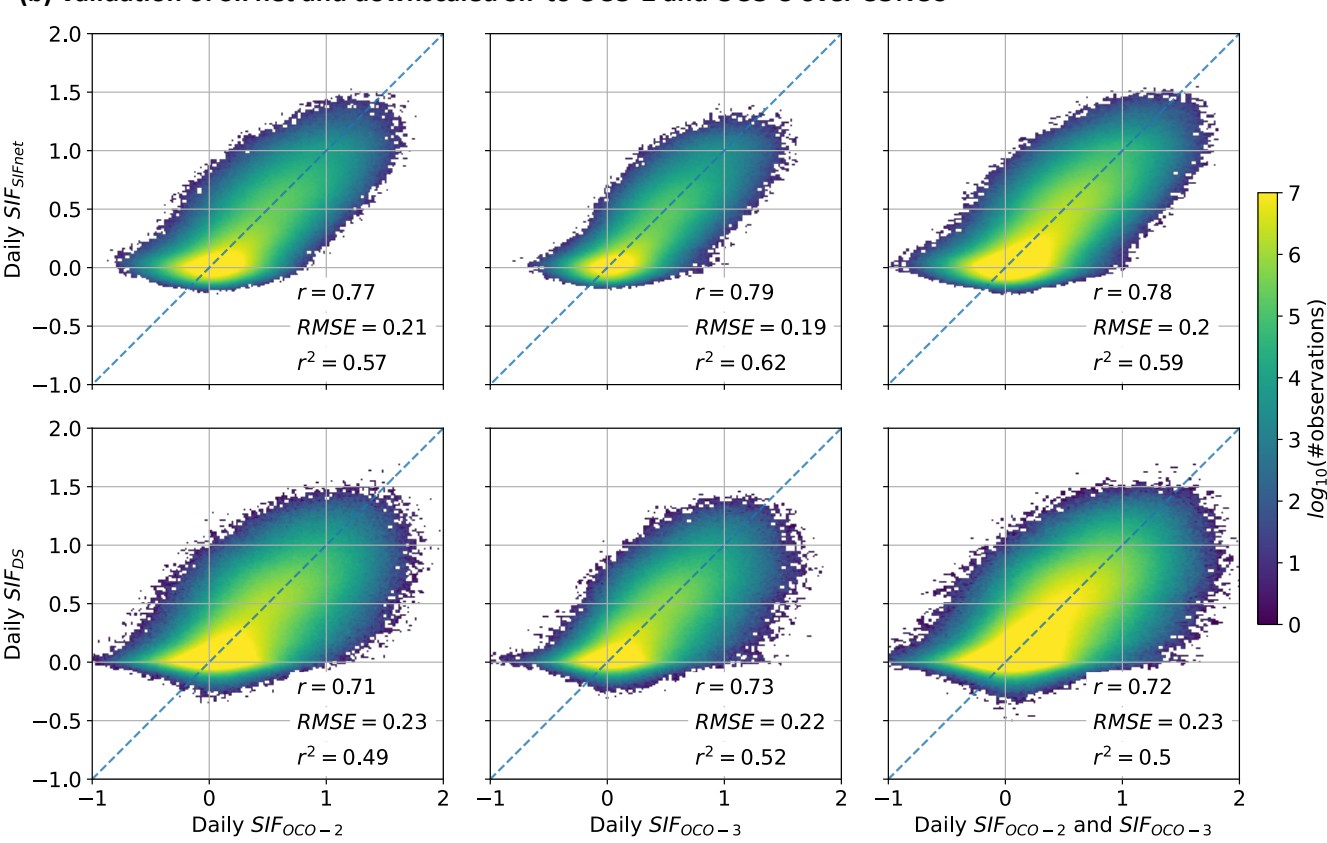

**Figure 7. Validation of SIFnet and Downscaled SIF to OCO-2 and OCO-3 SIF over CONUS.** Comparison from April 2018 until March 2021 in 16 day time steps. Daily OCO-2 and OCO-3 data is assigned to the closest 16 day time step. (a) shows the gridded correlation of the two products against the combined data of OCO-2 and OCO-3. We first use compute the SIF data from SIFnet and downscaling estimate that falls into the OCO footprint. Then we assign every OCO footprint to the closest grid point on the 1° grid dependent on the center location of that footprint and compute the Pearson correlation coefficient. (b) shows the scatter comparison of the weighted average of all grid cells on the 0.005° estimated SIFnet SIF (ours) and downscaled SIF (Turner et al., 2020) that fall into the OCO-2 or OCO-3 footprint.





Figure 8 presents a detailed comparison of SIFnet and the downscaled SIF in four US cities. The first column shows the SIFnet estimate, the second the downscaled SIF from Turner et al. (2020), the third the difference between SIFnet and down-

scaled SIF, and the last column the difference in correlation of the high resolution SIF estimates to combined OCO-2 and OCO-3 data on a 0.02° grid multiplied with the $L_1$-norm between the SIFnet and the downscaled SIF. The column on the right highlights regions where both the differences in predicted SIF are large and which product is performing better. As such, the right column will show white in areas where the difference in predicted SIF is small or the correlation with OCO is similar. While we observe large differences in predicted SIF for the urban areas (column 3), we don't find one product to perform sys-

tematically better in urban areas. This likely indicates the complexity in the SIF signal arising from urban areas. Additionally, urban areas make up a small fraction of the overall land mass and, as such, do not represent a large share of the training data in SIFnet. These factors likely contribute to the heterogenous performance observed in the right column of Fig 8.



Figure 8. SIFnet and Downscaled SIF, the Difference between these, and the Difference in Correlation to OCO-2 and OCO-3 for Four Urban Regions. First column shows the SIFnet estimate, second the downscaled SIF from Turner et al. (2020), third the difference between SIFnet and downscaled SIF and the last the difference in correlation of the high resolution SIF estimates to combined OCO-2 and OCO-3 data on a 0.02° grid multiplied with the $\ell^1$-norm between the SIFnet and downscaled estimate.





However, there are some notbale successes of SIFnet in urban areas that can be mapped directly to features in the urban area. Figure 9 shows both SIFnet and the downscaled SIF along with $NIR_v$ from MODIS and a true color image of Chicago.

A feature clearly stands out in both the downscaled SIF and $NIR_v$ image. This is a region with missing $NIR_v$ and effectively no downscaled SIF. However, SIFnet does not show a strong gradient here. This region corresponds to the Chicago airport. In the MODIS $NIR_v$ image it is visible that there is no valid data available for that region for the three investigated years. The downscaling method from Turner et al. (2020) relies only on $NIR_v$ in the weighting function. If there is no data available the region it is interpolated in space and time. Here, it shows that the method seem to fail in urban regions where no MODIS $NIR_v$

signal is available. SIFnet handles this region better and seems to rely on other auxiliary data if there is no MODIS $NIR_v$ available. In Figure 8 it is also visible that the SIFnet estimate correlates better with the OCO-X data than the downscaled SIF for the region of the Chicago airport.

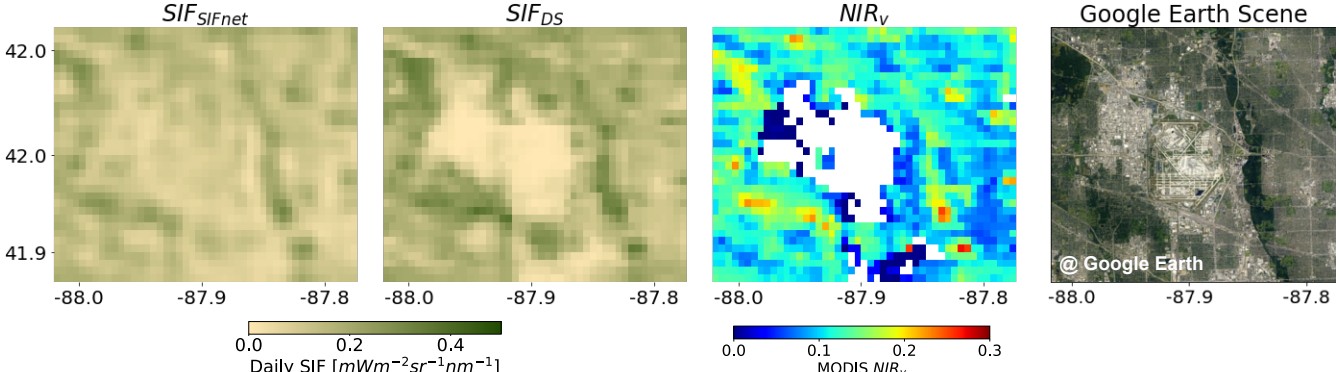

**Figure 9. SIFnet SIF, Downscaled SIF, MODIS NIR$_v$, and a Google Earth Cut-Out for a part of Chicago** Left panel shows the SIFnet estimate, second panel shows the downscaled estimate from Turner et al. (2020), third panel shows MODIS NIR$_v$ for Chicago, and last column shows the © Google Earth cut-out (Google LLC, 2021). For panel 1-3 the average data for April 2018 until March 2021 is shown.

## 5    Conclusions

Here, we develop a convolutional neural network (CNN) model named SIFnet to increase the resolution of TROPOMI SIF by

a factor of 10. The novelty of our method consists of using coarse resolution SIF measurements together with high resolution auxiliary data as model input to estimate high resolution SIF. After optimization and hyperparameter tuning of SIFnet, the estimated SIF at 500 m resolution yields an $r^2$ and $RMSE$ of 0.92 and 0.17, respectively, when comared against validation data (Figure 4). We further compare the output of SIFnet against a recently developed downscaling method to estimate high-resolution SIF (Turner et al., 2020) and evaluate both methods against independent observations from the Orbiting Carbon

Observatory 2 and 3 (OCO-2/3). SIFnet is found to perform systematically better than the downscaling approach when comared against indepdent measurements. Through interpretable machine learning methods, we identify the key features that SIFnet

 

utilizes to accurately predict high-resolution spatial patterns of SIF. We find that SIFnet relies heavily on the low resolution SIF feature ($SIF_{LR}$) and the vegetation index $NIR_v$ (Figure 5).

SIFnet is a multi-layer CNN that increases the spatial resolution of the TROPOMI SIF by a factor of 10. Our model uses

auxiliary datasets related to gross primary productivity and SIF as inputs and yields a high-resolution SIF estimate. The model is trained using three years of data from Asia, Europe, Africa, and South America. North America is used as the validation data set. Our loss function is comprised of two terms: the mean squared error and the structural dissimilarity index. The combination of these two terms improved the performance of our model.

SIFnet was further compared to the recent downscaled SIF product developed by Turner et al. (2020). The two high-

resolution estimates showed pronouced differences across the western US drylands. This difference is particularly interesting because these drylands tend to be low-productivity regions and traditionally have been difficult for SIF to accurately capture due to the low signal-to-noise ratio. Both high-resolution SIF estimates were compared to independent observations from OCO-2/3. SIFnet performed systematically better than the downscaled SIF ($r = 0.78$ for SIFnet, $r = 0.72$ for downscaling). SIFnet and the downscaling method also yielded differences in urban regions. However, there was substantial hetereogeneity

in the performance of SIFnet and downscaling in urban areas. One product did not perform systematically better than the other within urban areas. The mixed results in urban areas likely relates to both the complexity of the photosynthetic activity in urban areas as well as the lack of training data, as urban areas represent a small fraction of the total landmass.

We adapted techniques from the area of interpretable machine learning to assess the key features driving SIFnet. Specifically, we conducted random permutations to input datasets and assessed the impact on the resulting RMSE. From this, we found

that SIFnet relies most heavily on the low-resolution SIF feature ($SIF_{LR}$). The second most important factor is the MODIS vegetation index $NIR_v$. $NIR_v$ is also found to outperform the recently proposed $kNDVI$ vegetation index, in contrast to Camps-Valls et al. (2021). The interpretable machine learning approach also allowed us to spatial regions of importance for the different parameters. Interestingly, SIFnet relies more heavily on $NIR_v$ in the western drylands where the SIF signal-to-noise ratio is low. This implies that SIFnet is picking up on key physics that leads to the improved performance relative to the

downscaling method. Urban areas represent a region where SIFnet and the downscaling method performed comparatively well. Urban areas, while important in carbon cycle, represent a small percent of the total land mass. This means there is relatively less training data for SIFnet in urban areas. Further investigation into the processes controlling high-resolution GPP and SIF in urban areas is warranted. Overall, SIFnet represents a robust method to infer continous high spatial resolution information about processes related to gross primary productivity.

*Data availability.*   The high resolution SIF data is temporally available here: https://syncandshare.lrz.de/getlink/fiXHgheNbMsbMBDeWVV3DpzR/. It will be moved to a permanent data storage after acceptance.





*Author contributions.* JG, AJT, and JC conceived the study. JG compiled data sets, conducted data analysis, and generated figures. JG, AJT, and JC wrote the manuscript with feedback from all authors. JG did the literature research. JC provided project guidance. All authors contributed to the discussion. All authors have read and agreed to the published version of the manuscript.

*Competing interests.* The authors declare that they have no known competing financial interests or personal relationships that could have appeared to influence the work reported in this paper.

*Acknowledgements.* We thank Xiaojing Tang, Luca Lloyd, and Lucy Hutyra from Boston University, USA for providing us their valuable global data about forest fragmentation. JG and JC are supported in part by the Technical University of Munich-Institute for Advanced Study through the German Excellence Initiative and the European Union Seventh Framework Program under the Grant Nr. 291763 and, in part, 335 by the German Research Foundation (DFG) under the Grant Nr. 419317138. AJT was supported through the NASA Early Career Faculty program (grant 80NSSC21K1808) and the NASA Carbon Cycle Science program (grant 80HQTR21T0101).



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
