# Peer review of "A Convolutional Neural Network for Spatial Downscaling of Satellite-Based Solar-Induced Chlorophyll Fluorescence (SIFnet)"

_Biogeosciences, 2021_

## Author Comment (AC2)

**Response to Reviewer #1:**

Dear reviewer #1,

We are very grateful to your very helpful and insightful suggestions which have significantly improved the manuscript.
The modifications made during this revision can be seen in the **appended 'track change' version** of the revised manuscript. Our responses (in black) to the comments/suggestions (in blue) please find below. We also included the major changes in the manuscript to address the comments (in the form of tables).

Sincerely,

Johannes Gensheimer on behalf of all co-authors
* * *
**Reviewer #1 comments:**

The manuscript "A Convolutional Neural Network for Spatial Downscaling of Satellite-Based Solar-Induced Chlorophyll Fluorescence (SIFnet)" describes a new, downscaled SIF data set based on TROPOMI data. The downscalling is done with a Convolution Neural Network utilising a large number of variables in its feature space, including the coarse resolution SIF data itself. I am broadly familiar with the Machine Learning techniques in this paper, but I am certainly not an expert, so there could be issues that I am unaware of. However, to the extent of my knowledge everything appears to be rigorously constructed. Overall. the manuscript is well written and I only have some relatively minor comments.
We appreciate the reviewer's comments and feedback.

Minor comments and typos:
Thank you for all the comments! We have worked on all of them and marked them with the symbol ✓ after resolving. In some cases we have added further explanations or mention the changes in the manuscript.

✓ Paragraph starting L40 - I am not in complete agreement that making the spatial resolution of an instrument finer is the same thing as "improving" it. I appreciate many applications do benefit from finer resolutions than that of OCO2 SIF, but some applications benefit from coarse spatial resolutions (and there can be instrumental adavtages to this too). I also note that the spatial sampling of the two instruments is overlooked here, and that is arguably an even bigger challenge for integrating OCO SIF into ecological measurements.
That is correct and we agree with the statement of the reviewer. We have changed the term "improved" to "higher". We think that "higher" does not judge the resolution on being good or bad but just points out the difference. Thank you for this suggestion!

L. 44: improved -> higher

✓ At several points the phrase "ground truth" is used, but I think this is misleading as there is no actual "ground truth" in this paper. For example on Line 135, I would use the phrase "target variable" rather than ground truth.
It is indeed true that we do not have a ground truth measurement at the target resolution of this paper. For training on the other hand, we coarsen our input SIF to 0.5 degree and train on the actual TROPOMI gridded data at 0.05 degree. In this case we do have a ground truth. We have changed the ground truth term in L. 178-179 (after Equation 5) to target variable. The sentence in L. 133 (first sentence of section 3.1) just generally explains about CNNs. Here, we think that the term ground truth is suitable as we are not talking about our approach at this stage but introduce to the issue that we indeed do not have matching feature and ground truth data pairs. We have added the reference of (Bishop, 2014) to this sentence to point out that this is a general statement about CNNs.

✓ L36 the Sellers reference here should come after the statement about VIs representing photosynthetic capacity, not at the end of the sentence. (Because that paper doesn't mention SIF).

✓ L44 improves->improved

✓ L67 trains -> trained

✓ L67 CNN, should be defined here as it's the first usage in the main text.

✓ L68 uses -> used

✓ L107 photosynthetic -> photosynthetically

✓ L111 "channel" is the wrong word here. Maybe "band"?

✓ L128 the information about PCA presented here is repeated further down. I suggest deleting it here.

✓ L136 being -> begin

✓ L136 used -> used it

✓ L147 repeats information about PCA. It is probably better HERE.
Previous part is removed.

✓ L165 During training -> During training,

✓ Eqn 5 The MSE part needs a square (or it is just mean error, not mean squared error) and the DSSIM part has unbalanced parentheses.

✓ L199 did yielded -> yielded

✓ L224 "found SIF to perform poorly in the western US drylands" - I think some context is needed to make the meaning clear - how did SIF perform poorly? As a proxy for GPP? Or was the SNR bad?
Thank you for this suggestion. We clarified the context with a more detailed description in line 229.

| Recent work from Wang et al. (2021) found SIF to perform poorly in the western US drylands. | Wang et al. (2021) concluded SIF and $NIR_v$ to capture complementary events in western US drylands as a proxy for GPP and that the linear correlation of SIF to GPP was substantially lower in these regions compared to other vegetation types. |
|---|---|

✓ Eqn 6 - presumably the τ on the top of the fraction should not have a supersript "s" i.e. SZA(τ,x,y), because the intention is to integrate across the day?
Totally true. We have removed the s on top of the fraction. Thank you for pointing that out!

✓ L271 L1 is written differently in the main text and the figure caption. They should be consistent (but either is OK in my view).
Again, thank you for pointing that out! We have changed both to the superscript notation.

✓ L278 notbale -> notable

✓ L320 I suggest deleting the two sentences about urban areas here. The first sentence appears to contradict what was written in the previous paragraph. If the points about GPP are something the authors want to retain, they could be moved up into the earlier paragraph.
We have removed the part about urban areas in this paragraph, as we also think that the previous paragraph points out all important aspects we found on urban areas.

---

## Author Comment (AC3)

**Response to Reviewer #2:**

Dear reviewer #2,

We are very grateful to your very helpful and insightful suggestions which have significantly improved the manuscript.
The modifications made during this revision can be seen in the **appended 'track change' version** of the revised manuscript. Our responses (in black) to the comments/suggestions (in blue) please find below. We also included the major changes in the manuscript to address the comments (in the form of tables).

Sincerely,

Johannes Gensheimer on behalf of all co-authors
* * *
**Reviewer #2 comments:**

This manuscript presents a downscaled SIF product 'SIFnet' (x10 improvement to 500m) based on combining TROPOMI native low resolution gridded SIF, climate products, and remotely sensed gridded products using a convolutional neural network approach. The method development and product verification are well laid out and justified, with a comparison to a recent downscaled SIF product nicely incorporated. I am not a ML expert, so cannot comment on the specifics of the algorithm chosen, apart from that a substantial and commendable effort was made by the authors to: check model sensitivity to the input parameters (including developing a feature importance metric), overfitting, multicollinearity, validation, and intercomparison to other products.

We appreciate the reviewer's comments and feedback.

My main suggestions are to: i) incorporate more information on data filtering and quality assurance, including data coverage and missing values and their impact on final product values, ii) incorporating a more physiological perspective on what SIF is, and outline the implications of how choices in the methodology affect the interpretation of SIFnet SIF, iii) include a more detailed explanation for the apparent low correlation of TROPOMI SIF with OCO SIF and the implications for those using SIF products for GPP proxies, and iv) detailing why solar radiation at surface level was not included as an input as one of the main drivers of SIF and APAR (i.e. taking into account clouds). These general comments and more specific minor comments are found below.

General comments

Data filtering: the information provided is severely lacking. First, be specific about what data filtering methods and thresholds were applied for each dataset, and then specify the amount of data that was filtered (or available) after filtering. This is important on many fronts, especially as filtering will be non-uniformly distributed across space and time (e.g. cloudiness in the tropics, snow affected regions in winter etc.). Also, there will be a strong bias towards clear sky conditions as the reflectance indices won't produce valid data with clouds, yet SIF may still be retrievable through some clouds.

We have added the following part to the manuscript (L.163-171):

During training, all missing values in the data are set to zero. That mainly affects water regions, as the share of missing values in the used SIF data is caused by 91.2% by water. In case there is a missing value in the SIF training sample, all feature values of this pixel are also set to zero to ensure the network does not learn false relationships between the predictors and the target variable. For the MODIS bands we applied the quality index value 0 (best quality only). This filtering also removes pixels that include clouds. To ensure a high coverage we interpolated in time for MODIS. Further, training and test folds are selected based on coverage, i.e. the regions near the equation (between ±22.5°) are not included in the data cubes as MODIS reflectance is sensitive to clouds which appear frequently in these regions (compare Supplemental Figure S2 and S4). All static variables have full coverage on land. ERA5 data has full spatial and temporal coverage. We did not apply any further quality filtering on SMAP soil moisture data. The data is provided as a Level 3 product on Google Earth Engine.

Physiology perspective: Missing a more detailed explanation of what SIF is. Also, a physiology perspective (on the implications of method choices) beyond the intro is lacking seeing the emphasis of the downscaled SIF product is to monitor veg dynamics (e.g. GPP).

We have added the following part to the manuscript (L.35-39):

The electromagnetic signal SIF is emitted by chlorophylls during photosynthesis. SIF is emitted in the red-far-red wavelengths of 650-850 nm (Magney et al. 2020). It is a way, beside photochemistry and nonphotochemical quenshing, for de-excitement of the chlorophylls (Köhler et al. 2018, Turner et al. 2020). Even though the link between chlorophyll fluorescence and photosynthesis is nonlinear at leave and canopy scale (Magney et al. 2020), it does not hold for satellite scales, where a linear relationship of SIF to GPP is frequently reported (e.g., Turner et al. 2021, Magney et al. 2020, Frankenberg et al. 2011, Joiner et al. 2011).

TROPOMO vs OCO SIF: Can you explain why the r2 values are reasonably low (0.56-0.61) for TROPOMI vs OCO SIF? (Fig. S19 in S7)? This is important because they are both a measurement of the same metric and has implications for interpreting SIF. Is this purely time of day mismatch and spatial heterogeneity? This is especially important for putting into context the r2 values achieved in this study. Is the difference due to anything physiologically related, i.e. is OCO SIF more sensitive to GPP than TROPOMI (I cannot see why, but maybe the retrieval algorithms play a role)? A detailed comparison will be beyond the scope of this

paper, but a figure and some stats and more explanation are warranted seeing OCO SIF is used as verification.

We have added a figure to compare the daily corrected SIF of TROPOMI and OCO-2/3 (Supplemental Figure S19). Here, the r2 correlation coefficient increased from 0.56 to 0.61 and 0.61 to 0.62 between TROPOMI SIF and OCO-2 and OCO-3 SIF, respectively. Indeed, one might expect better correlations here as both present SIF at ~740 nm. However, as pointed out in Koehler et al. 2018, the uncertainty of both TROPOMI and OCO-2 SIF are expected to lead to a certain spread between the data sets. In addition, we do not account for differences in acquisition times and viewing-illumination geometry, which can lead to additional uncertainties in this comparison. For reference, when comparing single footprints of TROPOMI SIF to aggregated OCO-2 SIF for June 2018 globally, Koehler et al. 2018 found a r2 of 0.67, only additional aggregation lead to a r2 of 0.88. The mean deviation of TROPOMI SIF to OCO-2 SIF is close to the average standard deviation of TROPOMI SIF ($\sim 0.4\ mW \cdot m^{-2} \cdot sr^{-1} \cdot nm^{-1}$). In our analysis, from the 16 day product from TROPOMI SIF for April 2018 until March 2021 at 0.05 degree, we observe an average error in the TROPOMI SIF of $0.43\ mW \cdot m^{-2} \cdot sr^{-1} \cdot nm^{-1}$ for the CONUS. That error is close to the RMSE between instantaneous TROPOMI SIF and instantaneous OCO-2 SIF ($0.37\ mW \cdot m^{-2} \cdot sr^{-1} \cdot nm^{-1}$). To compare TROPOMI and OCO-2/3 SIF we aggregate the OCO-2/3 footprints to the same grid as our TROPOMI data (0.05 degree). As we aggregate multiple OCO-2 or OCO-3 footprints to match one TROPOMI grid cell at 0.05 degree the certainty of the OCO measurements increases, and therefore the RMSE between TROPOMI and OCO measurements decreases.

The explanation above is also included as a paragraph in the supplemental material (section S7)

Inputs: why not use incoming solar radiation at surface level from a gridded climate product rather than a clear sky flux equivalent? This would factor in cloudiness and is closer to the first driver of SIF magnitude in veg (PAR or APAR).

Thank you for this great suggestion! Indeed, PAR does strongly influence SIF and therefore we did a test to include the two variables "surface_net_solar_radiation" and "surface_solar_radiation_downwards" from ERA5 land (https://developers.google.com/earth-engine/datasets/catalog/ECMWF_ERA5_LAND_HOURLY) as features into our model. The model does not rely on it as shown in following feature importance figure (feature cluster SWin, marked by red circle):

[Figure]

A value of 1 means that the error of the model output with shuffled features is the same as with unshuffled features. That means that the model does not rely on the features, as the value does not influence the model output.

We think that this is due to the following reasons. The spatial resolution of available shortwave incoming radiation products is coarser than our prediction resolution during the training process (solar radiation variables of ERA5 land are at 11132 meters and our model output during training at 0.05 degree). Therefore several pixels at the resolution of 0.05 degree have the same value in shortwave radiation, but different values in the target variable SIF. Further, shortwave incoming radiation has (compared to other variables) a weak spatial gradient. Low resolution SIF that is ten times coarser than the model output is part of the feature space. As PAR strongly influences SIF and due to the weak spatial gradients of PAR the information of it is included in the SIF features.

Due to the weak influence of the two variables "surface_net_solar_radiation" and "surface_solar_radiation_downwards" from ERA5 land in on the model output which might be due to reasons listed in the previous paragraph, we propose to not include SWin in the feature space of our analysis.

Minor

Thank you for all the comments! We have worked on all of them and marked them with the symbol ✓ after resolving. In some cases we have added further explanations or mention the changes in the manuscript.

✓ Figure 1: what function was fit for each subplot to get the goodness of fit? Add info to figure caption.

We have added the following sentence to the caption of Figure 1.

> To quantify the goodness of fit we compute the Pearson correlation coefficient (r) for each subplot (Benesty et al., 2009).

✓ L 120 remove 'In' before Figure 2.

✓ L 123: any explanations offered as to why TROPIMI SIF and GOME-2 SIF show different behaviors of correlation with NIRv to Camps-Valls et a. (2021)?

Several aspects may contribute to the differences:.1) GOME-2 has a substantially lower spatial resolution, 2) the number of soundings of TROPOMI is much higher (factor of ~100), 3) differences in overpass time (morning vs midday), 4) differences in viewing-illumination geometry, 5) higher potential of cloud contamination of GOME-2 soundings due to the increased footprint size.

✓ L130: expand 'NN' (first usage)

✓ L136: check grammar of sentence beginning 'As such'

✓ L161: remove 'are' after 'we'

✓ L 166: perhaps explain the implication of setting missing values and to zero, especially over vegetated regions.

(same answer as to your comment on data filtering)

We have added the following part to the manuscript (L.163-171):

> During training, all missing values in the data are set to zero. That mainly affects water regions, as the share of missing values in the used SIF data is caused by 91.2% by water. In case there is a missing value in the SIF training sample, all feature values of this pixel are also set to zero to ensure the network does not learn false relationships between the predictors and the target variable (that also applies to vegetated regions). For the MODIS bands we applied the quality index value 0 (best quality only). To ensure a high coverage we interpolated in time for MODIS. Further, training and test folds are selected based on coverage, i.e. the regions near the equation (between ±22.5°) are not included in the data cubes as MODIS reflectance is sensitive to clouds which appear frequently in these regions (compare Supplemental Figure S2 and S4). All static variables have full coverage on land. ERA5 data has full spatial and temporal coverage. We did not apply any further quality filtering on SMAP soil moisture data. The data is provided as a Level 3 product on Google Earth Engine.

✓ Fig. 4: add SIF units to subplots

✓ L 199: remove 'did' before 'yielded'. Also a double full stop evident.

✓ L225: add space after 'notably,'

✓ L223: remove 'do' after 'approach'

✓ L276: both high res SIF products performing poorly in urban areas suggest they should be treated separately in a model and/or point to the requirement for even higher spatial res products (e.g. 5 or 10m).

Thank you for pointing this out! Yes, indeed there might be a need to treat urban areas differently. Potentially, we need to scale to higher resolutions (like you suggested) or focus the model training on urban areas. Optimizing the model on urban areas is an active research going on right now at our institute and out of scope of the current study.

Thank you for pointing this out, we modified the sentence.

| The interpretable machine learning approach also allowed us to spatial regions of importance for the different parameters. | The interpretable machine learning approach also allowed us to identify spatial regions of importance for the different parameters. |
|---|---|

✓ L322: repeated point of less training data – suggest removing or consolidating with the previous mention in the conclusion.

Supplementary Minor

✓ L29: remove 'is' after 'Australia'

✓ L51: see question mark

✓ Fig 6S: can remove one upper or lower diagonal as the plot is mirrored. Should subplots c & d have 1SD error bars seeing they represent a feature group?

We would like to keep the mirrored axis in the suplots a and b. In supplemental Figure S6 c and d, we show the explained variance of each principal component (PC). They do not represent feature groups but there is the same number of PCs as features. For the time variant case there are 19 features and therefore 19 PCs that each represents orthogonal axes and explain a certain share of the variance.

✓ L80: 'similar' to 'similarly' and be quantitative (provide a range).

✓ Fig S11c: can you comments on the mean value of rho_NIR compared to the equivalent plot in the main body for mean NIRv? Is the second feature (rho_NIR) more important than the second feature NIRv for the different input parameterisations? This point is worthy of another sentence or two in the main text.

We have added the following part to the manuscript (L.224-225):

[revised manuscript text omitted]
. Figs. S19a and S19b compare instantaneous SIF and Figs. S19c and S19d daily corrected SIF according to Equation 6 from the main text (Frankenberg et al., 2011). OCO SIF is gridded to a 0.05° grid by the center point of the pixel. Latitudinal borders are from 25° to 50° and longitudinal border are from -126° to -66.

135

Applying the daily correction to the SIF data, the $r^2$ correlation coefficient increased from 0.56 to 0.61 and 0.61 to 0.62 between TROPOMI SIF and OCO-2 and OCO-3  SIF, respectively. Indeed, one might expect better correlations here as both present SIF at 740 nm. However, as pointed out in Köhler et al. (2018), the uncertainty of both TROPOMI and OCO-2  SIF are expected to lead to a certain spread between the data sets. In addition, we do not account for differences in acquisition times and viewing-illumination geometry, which can lead to additional uncertainties in this comparison. For reference, when comparing single footprints of TROPOMI SIF to aggregated OCO-2 SIF for June 2018 globally, Köhler et al. (2018) found a $r^2$  relative to the value range.  of 0.67, only additional aggregation lead to a $r^2$ of 0.88. The mean deviation of TROPOMI SIF to OCO-2 SIF is close to the average standard deviation of TROPOMI SIF ( 0.4 $mWm^{-2}sr^{-1}nm^{-1}$). In our analysis, from the 16 day product from TROPOMI SIF for April 2018 until March 2021 at 0.05°, we observe an average error in the TROPOMI SIF of 0.43  $mWm^{-2}sr^{-1}nm^{-1}$ for the CONUS. That error is close to the RMSE between instantaneous TROPOMI SIF and instantaneous OCO-2  SIF (0.37 $mWm^{-2}sr^{-1}nm^{-1}$). To compare TROPOMI and OCO-2/3 SIF we aggregate the OCO-2/3 footprints to the same grid as our TROPOMI data (0.05°). As we aggregate multiple OCO-2 or OCO-3  footprints to match one TROPOMI grid cell at 0.05° the certainty of the OCO measurements increases, and therefore the RMSE between TROPOMI and OCO measurements decreases.

[revised manuscript text omitted]

---

## Author Response (AR1)

Dear Martin De Kauwe,

Thank you for your fast editing of our revised manuscript. We have included point-by-point answers to your three points of minor revision. We also included the major changes in the manuscript to address the comments (in the form of boxes).

Sincerely,

Johannes Gensheimer on behalf of all co-authors
* * *
You wrote "leave" and canopy scale, it should be "leaf"
Thank you for pointing out that major typo. We have changed it accordingly.

R2 asked for more details on the "the implications of method choices", I didn't see this in your revision?
We think that the model implications best fit in the section of feature importance (section 3.4). We have added the following part (starting in L. 213).

> The CNN is a data driven method and is not restricted by LUE terms. Although SM and meteorology (air temperature and precipitation) play a key role for photosynthesis, we find that they are not important to our model output. This does not necessarily imply that SIF is not linked to these parameters. This can be explained by 1) the variables SM and those from ERA-5 are at coarser resolution than the actual model output of the training phase which is at 0.05° (10000 m and 11132 m for SM and ERA-5, respectively). Therefore not each pixel at the resolution of 0.05° has its unique value for SM or ERA-5, but multiple cells can be within one SM or ERA-5 pixel. 2) Not only the auxiliary data of the model estimates higher resolution SIF, but it is computed together with coarse resolution SIF. Therefore, events like heat stress that impact a bigger area than the actual model output might be represented in the coarse resolution SIF. 3) We have aggregated the used data to 16 day time steps. LUE parameters influencing SIF might have a bigger impact on the estimation at higher temporal resolutions.

R2 also asked you to explain further about the "reasonably low R2", which you did, and noted something in the supplement. I feel like this commentary could also be added to the main manuscript for the interested reader; I'd like you to please consider this in revision.
Indeed it might be good for the reader to have it in the main text. It also helps interpreting the results of the model output. We have added the following part to the manuscript (starting in L. 286).

> Deviations between TROPOMI and OCO-2/3 also appear at a grid of 0.05° (Supplemental Figure~S19). The r2 coefficient is 0.61 and 0.62 between TROPOMI and OCO-2 and OCO-3 SIF, respectively. Indeed, one might expect better correlations here as both present SIF at ~740 nm. However, as pointed out in Köhler et al. 2018, the uncertainty of both TROPOMI and OCO-2 SIF are expected to lead to a certain spread between the data sets. In addition, we do not account for differences in acquisition times and viewing-illumination geometry, which can lead to additional uncertainties in this comparison. For reference, when comparing single footprints of TROPOMI SIF to aggregated OCO-2 SIF for June 2018 globally, Köhler et al. 2018 found a r2 of 0.67, only

additional aggregation leads to a r2 of 0.88. The mean deviation of TROPOMI SIF to OCO-2 SIF is close to the average standard deviation of TROPOMI SIF (~0.4 mW·m−2·sr−1·nm−1). In our analysis, from the 16 day product from TROPOMI SIF for April 2018 until March 2021 at 0.05°, we observe an average error in the TROPOMI SIF of 0.43 mW·m−2·sr−1·nm−1 for the CONUS. That error is close to the RMSE between instantaneous TROPOMI SIF and instantaneous OCO-2 SIF (0.37 mW·m−2·sr−1·nm−1). To compare TROPOMI and OCO-2/3 SIF we aggregate the OCO-2/3 footprints to the same grid as our TROPOMI data (0.05°). As we aggregate multiple OCO-2 or OCO-3 footprints to match one TROPOMI grid cell at 0.05° the certainty of the OCO measurements increases, and therefore the RMSE between TROPOMI and OCO SIF decreases.